# Identification of Multiple Replication Stages and Origins in the Nucleopolyhedrovirus of *Anticarsia gemmatalis*

**DOI:** 10.3390/v11070648

**Published:** 2019-07-15

**Authors:** Solange A.B. Miele, Carolina S. Cerrudo, Cintia N. Parsza, María Victoria Nugnes, Diego L. Mengual Gómez, Mariano N. Belaich, P. Daniel Ghiringhelli

**Affiliations:** 1Laboratorio de Ingeniería Genética y Biología Celular y Molecular-Área Virosis de Insectos, Instituto de Microbiología Básica y Aplicada (IMBA), Universidad Nacional de Quilmes, CONICET, Bernal B1876BXD, Argentina; 2Institute for Integrative Biology of the Cell (I2BC), Evolution and Maintenance of Circular Chromosomes, CEA, CNRS, Univ. Paris Sud, Université Paris-Saclay, 91190 Saint-Aubin, France; 3Laboratorio de Oncología Molecular, Universidad Nacional de Quilmes, CONICET, Bernal B1876BXD, Argentina

**Keywords:** baculovirus, AgMNPV, replication

## Abstract

To understand the mechanism of replication used by baculoviruses, it is essential to describe all the factors involved, including virus and host proteins and the sequences where DNA synthesis starts. A lot of work on this topic has been done, but there is still confusion in defining what sequence/s act in such functions, and the mechanism of replication is not very well understood. In this work, we performed an AgMNPV replication kinetics into the susceptible UFL-Ag-286 cells to estimate viral genome synthesis rates. We found that the viral DNA exponentially increases in two different phases that are temporally separated by an interval of 5 h, probably suggesting the occurrence of two different mechanisms of replication. Then, we prepared a plasmid library containing virus fragments (0.5–2 kbp), which were transfected and infected with AgMNPV in UFL-Ag-286 cells. We identified 12 virus fragments which acted as origins of replication (ORI). Those fragments are in close proximity to core genes. This association to the core genome would ensure vertical transmission of ORIs. We also predict the presence of common structures on those fragments that probably recruit the replication machinery, a structure also present in previously reported ORIs in baculoviruses.

## 1. Introduction

Baculoviruses are rod-shaped enveloped viruses with double-stranded-DNA circular genomes from 81,775 (Neodiprion lecontei NPV (NeleNPV)) to 178,733 kbp (Xestia C-nigrum GV (XecnGV)) [1]. Baculovirus infection produces two virion phenotypes: Occlusion derived virions (ODVs) and budded virions (BVs) [1]. The BVs appear during the initial stage of the multiplication cycle and are responsible for systemic infection of the host, while the ODVs are produced at the final stages [2]. This phenotype is so named because it is occluded in a protein matrix mainly based on polyhedrin or granulin (denominated occlusion bodies—OBs) that protects virions from different environmental injuries. ODVs are the agents responsible for primary oral infection in a new host [3]. From a biotechnological point of view, baculoviruses have many interesting features: They can carry large and multiple DNA inserts (at least 38 kbp) [4,5,6], and they can be easily produced and purified at high titers. These insect viruses (wild type species and recombinant variants) have been used in many applications, such as biological control agents against insect pests [7,8,9,10,11], protein expression systems [12,13], putative non-human viral vectors for gene delivery [14], vectors for vaccines [15,16], or as models of genome evolution [17,18], among others. The *Baculoviridae* family is divided into four genera: *Alphabaculovirus* and *Betabaculovirus*, comprising lepidopteran-specific viruses; *Gammabaculovirus*, which includes hymenopteran-specific baculoviruses; and finally, *Deltabaculovirus*, which, to date, comprises only Culex nigripalpus NPV (CuniNPV) and possibly other still undescribed or in the process of description dipteran-specific baculoviruses [1]. A notable feature of these pathogens shared by other viruses with large DNA genomes [19] is the existence of a different gene content among known species. However, they share a set of 38 orthologous sequences known as core genes [20,21], derived from ancestral baculovirus, while the other regions (each genome contain between 89 (NeleNPV) to 181 (XecnGV) open reading frames (ORFs)) are shared by various members or are unique to species [1,22]. This particularity complicates the studies conducted to understand the basic functions encoded in baculovirus genomes.

The core genes provide insights into some of the major functions required of all baculoviruses [22], may be defining what a baculovirus is. Core genes can be considered as ancestral essential genes that diverged in higher or lesser degrees [1]. However, it is not only ORFs that can be conserved among evolution as non-coding DNA sequences associated to regulation processes might also be conserved, hence a relation between core genes and those sequences might exist.

The baculovirus replication process has been extensively studied and similarities with other entities, such as herpesvirus, have been found [23], but many questions remain to be answered. Baculoviruses replicate in cell nuclei, probably using a combination of host and virally encoded proteins [22], but based on the size of the short-lived replication intermediates, the DNA synthesis would proceed by a markedly different mechanism to the host nuclear DNA [24], or the one used by other viruses, such as *Polyomaviridae* [25]. During replication, DNA molecules larger than the genome length are produced, which could be explained by two mechanisms: Rolling circle [26,27] or recombination-dependent replication [28,29]. In order to understand DNA replication in baculovirus infected cells, short-lived DNA replication intermediates have been characterized, identifying heterogeneous fragments up to 5000 nt, with an average size of 1000 to 2000 nt that could represent Okazaki fragments (Theta mechanism) or could also be fragments synthesized during recombination dependent replication [24].

A Theta or rolling circle mechanism needs an origin of replication (ORI), where a protein can bind to start the process by recruiting the replication machinery or by nicking DNA to allow strand displacement, respectively. In baculovirus, the replication seems to occur from multiple origins [22,23]. To identify this kind of sequence, transfection of plasmids containing viral DNA into virus infected cells has been used, subsequently digesting with DpnI to distinguish newly replicated DNA from input DNA [22]. With this approach, homologous regions (*hr*s) [30,31], non-*hr* sequences [32,33] and early promoters [34] have been identified as causing the infection-dependent replication of plasmids. Wu et al. [35] identified a different non-*hr* ORI inside a core gene of AcMNPV (*p143* or DNA helicase), broadening the spectrum of sequences involved in this function.

To date, different sequences have been described as origins of replication in baculoviruses. The homologous regions (*hr*s) from AcMNPV are 30 bp imperfect palindromes [36] that were originally postulated as viral origins because of their symmetric location, high AT content, and palindromic structure [37,38]. However, there are some baculovirus that do not have any *hr* [22], so *hr*s might not be required by all baculoviruses for replication. Furthermore, Carstens and Wu showed that no single *hr* is essential for replication in a cell culture [39]. On the other hand, the non-*hr* ORI found inside the fragment HindIII-K of the same virus [40] has no obvious similarity with any of the *hr* sequences. This led us to question if different regions can alternatively or simultaneously function as ORI, or if there is a common secondary structure that is recognized by the replication machinery. Besides, this region has been found in defective AcMNPV genomes, generated after extensive serial passage of the virus in an insect cell culture, suggesting an important role in replication [32]. It seems that the AcMNPV DNA replication system does not require *hr*s for initiation. In fact, virtually any sequences can be multiplied when plasmids and viral replication-essential genes are co-transfected into susceptible cell lines since the initiation of DNA synthesis could be related to the chromatin structure [34]. However, when given time, after plasmid transfection prior to infection, it has been observed that only the plasmids containing viral promoter or *hr*s replicate [23,34]. Following this strategy, the previously mentioned non-*hr* direct repetitions located into the coding region of the *p143* gene were functional as ORI in insect and mammalian cells [35].

In this work, an unbiased method on a genome-wide scale was developed to isolate origins of replication in the Anticarsia gemmatalis multiple nucleopolyhedrovirus (AgMNPV), the alphabaculovirus most used worldwide as a bioinsecticide to control crop pests [41]. We combined a genome library with a previously described transient replication assay [34,42]. Exhaustive analyses of recovered sequences obtained in this work and comparison with previous reports are discussed to understand how these entities multiply their genomes. We found three kinds of origins, according to their genome location: Homologous regions (ORI-hrs), intergenic regions (ORI-igrs), and intragenic regions (ORI-grs); particularly, we detected non-*hr* ORI positioned within or close to core genes.

To find out more about baculovirus replication, viral DNA replication kinetics were monitored post-infection into the susceptible UFL-Ag-286 cells hourly for 30 h. This assay identified four different phases. Two of them were exponential stages (the second and fourth ones) and the other two showed no significant increase in DNA synthesis (the first and the third ones).

Taken together, the results shown here indicate that baculoviruses can use different kinds of sequences to replicate their genomes and they can undertake different replication mechanisms. The common feature that allows the different ORIs to be recognized by the replication machinery remains to be answered. However, this work constitutes a step forward in understanding the viral cycle.

## 2. Materials and Methods

### 2.1. Cell and Virus Stocks

The insect cells UFL-Ag-286 [43] were grown at 27 °C in GRACE’s medium (Invitrogen, Carlsbad, CA, USA) containing 10% *v*/*v* fetal bovine serum (FBS; GBO, Kremsmünster, Austria) and supplied with antibiotics and antimycotics (Invitrogen). Initial stocks of AgMNPV-2D [44,45] were multiplied in *Anticarsia gemmatalis* (third instar larvae) by *per os* infection using OBs, and by the exposition of monolayers of UFL-Ag-286 cells with BVs in plastic tissue culture flasks. The virus stocks used in all experiments were tittered by plaque assay [46] in UFL-Ag-286 cells and maintained as culture supernatants.

### 2.2. Quantitative Real Time PCR Assays for AgMNPV and UFL-Ag-286 Cells

A molecular approach to quantify AgMNPV and UFL-Ag-286 cells by two quantitative real time PCRs (qRT-PCR) was developed. Thus, pairs of primers that bind in the virus *egt* gene (pFw-egt: 5′TAATCGGCAAACGCCTCTAC3′; pRev-egt: 5′CGATGCGGCAAACAAACAAC3′) and into the cell *actin* gene (pFw-Ac: 5′TAATCGGCAAACGCCTCTAC3′; pRev-Ac: 5′CGATGCGGCAAACAAACAAC3′) were synthesized. To design the above oligonucleotides, the AgMNPV genome (NC_008520.2) and our own data (KX139200.1) from the actin gene of *Anticarsia gemmatalis* were used. First, end-point-PCR assays were performed using standard conditions and Taq DNA pol (PB-L, Bernal, Argentina) to produce the *egt* and *actin* amplicons, which were then molecularly cloned in pGEM-T-Easy (Promega, Madison, WI, USA) and *Escherichia coli* Top 10 [47]. To verify the identity of both constructs (pGT-egt and pGT-act), sequencing reactions with the universal primers, T7 and Sp6, were carried out (Macrogen Services, Seoul, South Korea). After this, both qRT-PCR were optimized in sensitive and dynamic range parameters using serial dilutions of pGT-egt and pGT-act as templates (1 μL), 25 μL of 1× Maxima™ SYBR Green qPCR Master Mix (Fermentas, Waltham, MA, USA), 0.3 μM of every primer, and a Smart Cycler (Cepheid Inc, Sunnyvale, CA, USA) to cycling temperatures and detect fluorescence. Plasmid DNA was previously purified (Qiaprep Miniprep Kit; Qiagen, Hilden, Germany), quantified by spectroscopy (Nanodrop 1000; Thermo Fisher, MA, USA), and the copy number was determined using the Thermoscientific copy number calculation tool (http://www.thermoscientificbio.com/webtools/copynumber/). The temperature conditions optimized for both genes were: 10 min at 95 °C (1 cycle), 15 s at 95 °C, 30 s at 60 °C, and 30 s at 72 °C (40 cycles).

### 2.3. Replication Kinetics

To understand the replication process of AgMNPV multiplying on in vitro cell cultures, a kinetic assay was performed. To this, synchronized monolayers (70% confluence) of UFL-Ag-286 cells [45] growing on 24 multiwell plates were exposed with AgMNPV (multiplicity of infection—moi—10) for 1 h. Then, cells were washed three times with 500 µL of phosphate buffer saline (PBS) and finally incubated with growth medium (500 µL of GRACE´s with 10% *v*/*v* SFB) for 1 to 30 h at 27 °C. The whole conditioned medium and cells were recovered from wells every 1 h, collecting three samples of cells and supernatants (three wells) per time. The viral DNA (vDNA) was isolated from supernatants by the miniprep method [46], reserving 50 µL to infection assays. For the remainder of the sample, DNA (including vDNA and cellular DNA—gDNA) was isolated from cells by treatment with Proteinase K (Sigma-Aldrich, St. Louis, MO, USA) in lysis buffer (0.2 M Na_2_CO_3_, 0.02 M NaCl, 0.2 M EDTA, 0.5% *w*/*v* SDS), phenol-chloroform extracted, and subsequently precipitated with ethanol and 0.3 M NaOAc [47]. Finally, to quantify relative levels between vDNA and gDNA, qRT-PCR assays previously optimized were applied. Thus, all DNA samples were analyzed with 1× Maxima™ SYBR Green qPCR Master Mix (Fermentas, Waltham, MA, USA) into a Smart Cycler (Cepheid Inc, Sunnyvale, CA, USA) using as pairs of primers, pFw-Act/pRev-Act and pFw-egt/pRev-egt. To estimate the number of DNA molecules, the recombinant plasmids, pGT-egt and pGT-act, were used as calibrators. To normalize data, gDNA was calculated arbitrarily considering one copy of *actin* gene per cell. The results were plotted as quotients between vDNA and gDNA and the statistical error and ratios were calculated and then propagated. Unpaired T test was done for statistical analysis. The replication rates corresponding to exponential phases were estimated from the slopes of the curves using SigmaPlot software. To verify the presence of BVs in culture supernatants, 50 µL of each conditioned medium sample were used as inoculums on UFL-Ag-286 cells growing on 48 multiwell plates and incubated for 1 week at 27 °C. The presence or absence of infection was determined by microscopy (Nikon Eclipse TS100).

### 2.4. AgMNPV Genomic Library

A plasmid library of overlapping fragments derived from the AgMNPV genome was produced into the pBlueScript SK (+) vector (pBSK) (Stratagene, San Diego, CA, USA). Briefly, AgMNPV genome (vDNA) was isolated by the full scale method from BVs, which include concentration by ultracentrifugation, proteinase K (Sigma) treatment, phenol-chloroform extraction, and alcohol precipitation steps [47]. These virions were previously produced in controlled conditions in UFL-Ag-286 cells (flask of 175 cm^2^; 70% cell confluence and state of synchronization [45]; moi 0.1; 5 days of incubation). After verifying the quality by agarose gel electrophoresis and spectroscopy (Nanodrop 1000), vDNA was partially digested with HaeIII (Fermentas, Waltham, MA, USA) to obtain overlapping fragments ranging between 500 and 2000 bp, which were recovered from preparative electrophoresis using a Zymoclean Gel DNA recovery Kit (ZYMO Research, Irvine, CA, USA). Subsequently, vDNA fragments were ligated using T4 DNA ligase (Hoffmann-La Roche, Basel, Switzerland) to pBSK, which was previously treated with EcoRV (Fermentas, Waltham, MA, USA) and dephosphorylated (Shrimp Alkaline Phosphatase, Fermentas, Waltham, MA, USA). Finally, recombinant DNA was transformed by electroporation (BIORAD Gene Pulser II) in *Escherichia coli* Top 10 (Invitrogen, Carlsbad, CA, USA) and then 500 colonies white colored (in presence of X-Gal; Sigma) and resistant to ampicillin (Sigma-Aldrich, St. Louis, MO, USA) grown on solid Luria-Bertani (LB) plates (Britania, Buenos Aires, Argentina) were selected. The genome coverage of the AgMNPV library was estimated. For this, bacterial clones randomly selected (20%) were used to isolate recombinant plasmids with the goal to determine the insert length by treatment with restriction endonucleases and electrophoresis analyses. All clones were individually maintained at −80 °C in storage medium (LB broth supplied with 15% *v*/*v* glycerol).

### 2.5. Transient Replication Assay

All recombinant plasmids from the AgMNPV library were isolated by the standard alkaline method [47] from 5 mL cultures (LB broth supplied with ampicillin). Then, constructs were mixed in pools of 25 clones (selected without any bias and maintaining the same copy number of each one) and 2 µg of mixed DNA for each group were transfected using CellFectin Reagent (Invitrogen, Carlsbad, CA, USA) according to the manufacturer´s instructions on 8 × 10^5^ UFL-Ag-286 cells per well grown in standard condition in various six multiwell plates. As a control, some wells were transfected with 2 µg of pBSK without insert. After a period of 6 h, transfected cells were also infected with AgMNPV at moi 1 and subsequently incubated at 27 °C for 48 h, reserving some wells without virus exposure as controls [23,34]. At 48 h, cells from all wells were harvested and plasmid DNA was purified using the alkaline method [47]. The recovered DNAs were overnight treated with 2.5 U of DpnI (Fermentas, Waltham, MA, USA)—reserving some samples without enzyme hydrolysis as controls—to eliminate the DNA template synthesized in bacteria, cleaned up by phenol-chloroform extraction and alcohol precipitation, and transformed in *E. coli* Top10. Control samples without any plasmid (only UFL-Ag-286), or containing only the empty cloning vector (UFL-Ag-286/pBSK, UFL-Ag-286/pBSK digested with DpnI) were used to adjust the experimental conditions. The insert sequences of transiently replicated constructs were obtained by Sanger sequencing (Macrogen Inc, Seoul, South Korea) with universal primers (T3 and T7), and then compared against the AgMNPV-2D genome (NC_008520.2). The transient replication assay was carried out twice in independent events.

### 2.6. Bioinformatics Analysis

The sequences obtained from the virus genome library were identified by the BLAST algorithm (www.ncbi.nlm.nih.gov/blast) using the information of AgMNPV-2D (NC_008520.2). The AT content of the whole genome was calculated and profiled using an overlapped sliding window strategy (window = 1000 nucleotides, step size = 100 nucleotides), using a DNA base composition analysis tool developed by Jie Zheng in 2004 (http://molbiol-tools.ca/Jie_Zheng/). The average AT content of the whole genome and library inserts containing AgMNPV´s ORIs were also calculated with the same tool. Besides, the amount of A or T runs (4–8 mers) were dimensioned by a similar approach and profiled using an index, where the number of runs was normalized with the average value of the AgMNPV AT content. To detect putative promoters, the available information regarding typical early and late motifs described in *Baculoviridae* [48] and the DNA Pattern Find program [49] were applied. Moreover, to predict palindromes, the European Molecular Biology Open Software Suite (EMBOSS) Palindrome tool (http://emboss.bioinformatics.nl/cgi-bin/emboss/palindrome) was utilized [50]. After characterizing the ΔG_0_ of stem loops constituted by inverted repeats of different lengths (using RNAstructure Software Package) [51], the following search parameters were selected: Minimum length of palindrome, 5; maximum gap between repeated regions, 18; number of mismatches allowed, 1. DNA folding routine of Mfold web server [52] was used to predict the secondary structures of DNA with the following modified parameters: Folding temperature (27 °C) and maximum distance between the paired bases (40 bp).

## 3. Results

### 3.1. AgMNPV Kinetics of Replication

A kinetic analysis of virus DNA synthesis using in vitro cell cultures was performed, to contribute with experimental evidence about the AgMNPV genome’s replication that allow the identification of different phases and rates of synthesis. For this, a qRT-PCR approach and two DNA loci as templates were selected: The *egt* virus gene (vDNA) and cell *actin* gene (gDNA). A segment of the actin gene had been obtained using ad hoc designed universal primers in an end-point-PCR assay and subsequently sequenced (GenBank KX139200) since there were no available *Anticarsia gemmatalis* sequences in the databases. Thus, synchronized UFL-Ag-286 cells were exposed using an excess of AgMNPV to synchronize the infection, profusely washed to remove the virions that did not enter cells, and finally were incubated in growth conditions. Every 1 h-interval, gDNA and vDNA were measured (Figure 1).

Results revealed different stages of AgMNPV replication. The first stage finished at 5 h post virus exposure of infective particles (hpi) when the quantity of vDNA increased significantly with respect to the previous time (*p* < 0.0001). Then, a second phase between 5 and 16 hpi showed that vDNA progressed in an exponential tendency (µ_v_: 0.37 ± 0.027) without detecting DNA in the conditioned medium. After this time, BV production started because virus genomes were detected outside cells (corroborated by in vitro cell infection assays) and a decrease in the whole production of vDNA was observed (*p* < 0.0001). This is in accordance with the fact that many genome molecules were packaged and emerged as BVs, and in consequence, they could not have been used as new templates for replication. This particularity occurred for 5 h (from 17 hpi to 21 hpi; *p* < 0.0001), defining a third replication phase. Finally, the process progressed in another exponential phase of growth (µ_v_: 0.34 ± 0.037) until the end of the assay. Probably, a fifth phase might occur, where the vDNA quantity does not significantly vary over time, but this experiment did not arrive to such a plateau. These results are in accordance with previous results, where DNA replication started 5 to 6 h post infection and reached a maximum rate by 18 h post infection [53].

### 3.2. Discovering ORI Sequences in AgMNPV

As different kinds of sequences have been described as origins of replication in baculoviruses, the genome of AgMNPV was screened using a modified DpnI assay. Briefly, a virus genome library containing overlapped fragments between 500 and 2000 bp was constructed and then 500 clones, which represent three times the genome coverage, were evaluated in groups of 25 by two independent transient replication assays (Appendix A).

Once the plasmids from infected cells had been recovered after an incubation time of 48 h in growth conditions (period that includes all replication phases previously identified by the kinetic assay), DNA was treated with DpnI to remove the original constructs derived from *E. coli*. Then, DNA was transformed into bacteria to select the plasmids multiplied by the viral machinery inside the insect cells. From the growing colonies, we randomly chose 16 of them to identify the viral sequences that could function as putative ORIs (DNA inserts of the recovered library constructs) were determined by direct sequencing. Thus, 14 genome regions were found and were classified according to their locations in three main groups: ORI-hr (homologous regions); ORI-igr (intergenic regions located between two annotated ORFs that may also contain small annotated ORFs); and ORI-gr (gene regions corresponding with sequences immersed into some annotated ORFs) (Figure 2; Table 1).

As expected by previous reports in other species, two virus regions contained a complete *hr* (Ori-hr, pBS-AGN39 contains *hr*9 and pBS-AGN113 contains *hr*7), supporting the capability of these loci to recruit the replicative machinery.

For the other recovered sequences, it is important to note that most of the 14 virus segments containing putative ORIs are next to or within ORFs. Interestingly, they are located inside genes shared by all baculoviruses (core genes). Two plasmids (pBS-AGN33 and pBS-AGN162) contain the same insert in the same polarity and four clones (pBS-AGN104, pBS-AGN119, pBS-AGN131, and pBS-AGN141) carry sequences of the same genome locus, adding value to the methodological approach chosen to find putative ORIs.

Regarding the ORI-grs, clones pBS-AGN104/131, pBS-AGN136, pBS-AGN153, and pBS-AGN168 contained virus segments located inside of core genes (*vlf-1*, *lef-8*, *p40*, and *vp91*, respectively). By contrast, only pBS-AGN26 and pBS-AGN144 clones contained sequences (*egt* and *lef-3*) that do not belong to any core genes but correspond to genes that are important to all lepidopteran baculoviruses [20,21].

Plasmids named ORI-igrs contained sequences composed by the 5′ or 3′ extremities of coding sequences and the non-coding region immersed between ORFs. Plasmid pBS-AGN119/141 contained a fragment of *vlf-1* and ORF85 (*ac78-like*), both being core genes [20]. The clone pBS-AGN162/33 carried fragments of the core genes, ORF67 (*lef-5*) and ORF68 (*ac98-like* or *38k*) [21]. However, pBS-AGN27, pBS-AGN133, and pBS-AGN150 did not contain fragments of core genes but contained sequences of genes shared by all group I alphabaculoviruses, reflecting that they could be important for AcMNPV and its lineage (Table 1; for more information, see Appendix B).

### 3.3. Characterization of ORI Sequences from AgMNPV

Since sequences of dsDNA rich in AT content can be more easily unwound and thus act as starting points to replication, we analyzed the AT contents in all the virus segments recovered by the modified transient replication assay previously described. We found that only a few clones had a higher AT percentage value than the average percentage of the AgMNPV genome, so we could not establish any correlation between the AT percentage inside clones and their capability to replicate, but we identified that they all have 4 to 6mer runs of A or T (Figure 2). Particularly the ORI-hrs were not located in genome regions rich in A or T runs, but some ORI-grs (pBS-AGN136, pBS-AGN144) and ORI-igrs (pBS-AGN27, pBS-AGN133) were positioned in the loci where the highest concentration of runs occurred.

Other sequence features were also investigated. Since it was previously reported that early promoters could function as ORI, the existence of typical patterns was researched. The presence of palindromes was predicted, thinking that this kind of sequence may form secondary structures (Table 2).

The analysis of results showed that all segments contained early promoter patterns, but only a few were considered functional by its position relative to adjacent ORFs. As expected, ORI-igrs had a higher proportion of these motifs than the other ones (Table 2; for more information, see Appendix B).

Additionally, all sequences showed the presence of perfect and imperfect palindromes of different lengths. All detected ORIs had perfect palindromes of at least a 5 bp length. ORI-hrs and ORI-igrs revealed a greater number of perfect palindromes than ORI-grs (20.5 ± 13.4, 14.8 ± 8.6, and 5.3 ± 4.3, respectively). The imperfect palindromes found are seven to eight nucleotides in length. Moreover, imperfect palindromes containing more than 10 nucleotides were also detected in the three kinds of ORIs (for more information, see Appendix B).

As the AT content and the distribution of promoter motifs were not a shared property among all detected AgMNPV’s ORIs, the study continued with a focus on the secondary structure since all presented palindromes. Considering that this kind of sequence in double stranded-DNA can form cruciform structures, the virus regions recovered by the transient replication assay were explored to detect shared secondary structures. Notoriously, three hairpins that conform a unit, in some cases, repeated several times, were predicted in all ORIs from AgMNPV (Figure 3, Table 3). This common structure was slightly rich in AT content (50%–60%) and derived from DNA segments containing an average of 65 nucleotides (range of 39 o 111 bp) and could form three stems, where the first and third ones had an average of 4.6 nucleotides (ranges of 2–14 and 2–10, respectively), while the second one had 8.5 nucleotides (range: 5–16). The two linkers located between stems had an average of three nucleotides, ranging between 0 and 6. In all cases, the stem two was the longest one.

Considering that a similar structure was reported in 2014 for Sucra jujuba NPV’s *hr*s [54], we explored other baculovirus reported ORIs from different species. Thus, the *hr*s from AcMPNV [55], LdMNPV [56], SeMNPV [57], and non-*hr* ORIs from AcMNPV [42], SeMNPV [58], OpMNPV [59], and CrleGV [60] were studied. Surprisingly, the structure found in the AgMNPV’s ORIs was also detected in the *hr*s of AcMNPV (*hr*1, *hr*1a, *hr*2, *hr*3, *hr*4a, *hr*4b, and *hr*5), of LdMNPV (*hr*1, *hr*2, *hr*3a, *hr*3b, *hr*3c, *hr*4, *hr*5, *hr*6, *hr*7a, *hr*7d, and *hr*8) and SeMNPV (*hr*2, *hr*3, *hr*5, and *hr*6), and in the all non-*hr* repetitive sequences of AcMNPV, SeMNPV, OpMNPV, and CrleGV (Appendix A and Appendix A). In some cases, the linkers are longer than the ones detected in AgMNPV replicative sequences, as in LdMNPV (*hr*7a and *hr*7d), but the pattern is conserved. Interestingly, the AcMNPV´s *hr*1a is considered as sufficient for supporting plasmid replication [26] and could adopt a structure similar to stem two.

All analyses performed on the sequences from AgMNPV that acted as ORI were summarized using a Venn diagram (Figure 4). ORI-hrs are rich in AT content and A/T runs, they have palindromes, and can form a secondary structure; meanwhile, ORI-igrs have a similar value of AT content with respect to the genome average but share the other characteristics. Finally, ORI-grs seem to be rich in A/T runs and palindromes, also enabling the formation of the secondary structure previously described.

## 4. Discussion

To successfully carry out the replication of dsDNA genomes, multi-protein complexes are needed to distinguish the starting points of DNA synthesis (ORIs), promote replisome formation, and then progress along the whole replicon. This whole process has been extensively studied for prokaryote and eukaryote organisms. However, the study of viruses becomes more complicated since it involves viral and host proteins. Two models had been proposed for baculovirus initiation: (a) Replication is coordinated with packaging, initiated in multiple ORIs to greatly amplify the viral DNA that can be produced in a given time; (b) replication is independent of packaging and undergoes extensive recombination [22]. However, data published until date does not permit discernment between those two possibilities. Furthermore, these models do not consider the two stages in the baculovirus life cycle: BVs and ODVs production. It is known that in eukaryotes, multiple origins of replication are activated during the cell cycle in a program precisely coordinated in space and time [61].

The results obtained on AgMNPV in this work support the existence of different phases during baculovirus infection. In agreement with previous publications [62], we detected an early phase (phase one) in which BVs require about 5 h to enter the cell, migrate across the cytoplasm, enter into the nuclei, and hijack host transcription machinery to finally start replication. Then, a first exponential phase of replication occurs until the beginning of budding (phase two), an event that initiates a decrease in DNA productivity (phase three). Later, this stage is followed by phase four, characterized by another increase in genome synthesis. This behavior could be due to a change of the mechanism involved in DNA replication probably associated with the production of late virus proteins. DNA synthesis rates (μ_v_) were similar in the two exponential phases. Since part of the DNA that could be used for replication is constantly removed to be packed into BVs and ODVs during phase four, it is possible to say that a different process of replication is being carried out.

The different loci identified here could recruit the replication machinery at some point during the infection cycle and allow the different replication stages by using different kinds of origins. It has been discussed before that despite the mechanism of replication (Theta or Sigma), baculoviruses may utilize multiple origins with different levels of initiation efficiency, starting primarily at the *hr* origin [63].

Analyzing the results from the transient assay, we found that two of the plasmids able to replicate contained *hr*s sequences. Most baculoviruses have several *hr*s dispersed over the genome [22,33], varying from 3 in CrleGV [64] to 17 in SpltNPV-G2 [65] but cannot be considered core sequences. All group I alphabaculoviruses have *hr*s but not all group II species, such as ChchNPV [66] and TnSNPV [67]. Although *hr*s are not conserved in all baculoviruses, they have been detected as ORI as demonstrated by transfection replication assays in different species, including AcMNPV [39,68], SeMNPV [57], CfMNPV [69], and LdMNPV [56]. In AcMNPV, the study showed that plasmids containing *hr*s replicate 5- to 19-fold higher than those containing other genomic DNA fragments, and the replication rates were associated with the number of palindromes or their orientation [68]. However, recombinant viruses, where *hr*1, *hr*1a, *hr*2, *hr*3, *hr*4, or *hr*4b were knocked-out, revealed that none of *hr*s would be essential for DNA replication [39]. In our work, the selection of fragments was done without bias, thus the appearance of *hr* adds new evidence of its function as ORIs.

Furthermore, our data opens a new panorama regarding the kind of sequences that can be used as origins of replication in AgMNPV. Apart from *hr*s, other ORIs containing tandemly repeated sequences have been characterized in AcMNPV [39,42], SpliMNPV [70], SeMNPV [58,71,72], OpMNPV [33,59], and CrleGV [60]. Also, in AcMNPV, two other kinds of non-*hr* ORIs have been described: Early promoters [34], and a segment of the DNA helicase ORF [35]. In our work, 12 different sequences of AgMNPV classified as intergenic and intragenic regions were detected and named ORI-igrs and ORI-grs, according to the presence of putative functional early promoters or the presence of fragments of ORFs, respectively. In this work, one of the fragments (pBS-AGN150, classified as ORI-igr) contains a direct repetition (dr12, 95,016-95,057 nt). Why are the sequences found in AgMNPV able to initiate replication? Possible answers should include the role of IE-1 protein because both early promoters and *hr*s are sequences able to bind to that polypeptide [73,74]. For example, AcMNPV IE-1 binds to the *hr* palindromic sequences [75] and this interaction would be essential to function as transcriptional enhancers [76]. Besides, it has been proposed that the palindrome core of AcMNPV *hr* (28 bp) acts as ORI, whereas the flanking sequences are required for enhancer activity [68]. Thus, IE-1 could be an ORI binding protein or be necessary to transactivate all early genes involved in DNA replication [20]. As mentioned above, AdorGV, AgseGV, CpGV, CrleGV, PrGV, and SpliGV have no *hr*s, although they have *ie-1* orthologs [77,78,79]. Therefore, the recovery of *hr*s (pBS-AGN39, pBS-AGN113) and regions containing early promoters (pBS-AGN27, pBS-AGN119/141, pBS-AGN133, pBS-AGN141, pBS-AGN150, and pBS-AGN33/162) makes sense considering that IE-1 would be an ORI binding protein.

The reason why intragenic regions are able to initiate replication processes still remains to be answered. Probably, a common secondary structure might be the need factor recognized by the replication machinery to initiate DNA synthesis. In fact, there are reports in eukaryotic genomes about the presence of loops on mitochondrial ORI [80] as well as in circovirus [81] and herpesvirus ORIs [82]. It is possible that particular palindromic regions form a curve or cruciform structure that are recognized by the proteins responsible for the initiation of DNA synthesis [29,55]. As it is known, both transcription and replication require the local unwinding of double stranded-DNA. In these processes, the generated single stranded-DNA is used as a template for the synthesis of RNA transcripts or to produce daughter strands. Theoretically, the *hr*s palindromes may form cruciform structures although some evidences [55] indicate that this situation only occurs when the imperfect palindromes are modified to be perfect. However, the formation of a cruciform structure under certain conditions during viral replication or the stabilization by protein attachment or by particular neighboring sequences cannot be discarded. Again, IE-1 could play an important role in replication by recruiting a replication complex, since it has been shown that it can bind to *hr*s in their cruciform or extended shape [55]. In AcMNPV, a single palindrome repeat from an *hr* is a minimal requirement for plasmid dependent DNA replication [26], and a single palindrome could adopt a hairpin structure.

The presence of palindromes was detected in all the sequences found here. Thus, the DNA secondary structure could be the key feature recognized by the replication machinery in baculoviruses to initiate DNA synthesis and be positively selected in preference of a primary sequence. Many origins of replication contain inverted repeats, since the formation of a cruciform structure is energetically more favorable than simple DNA melting and can absorb part of the strand generated folding it into hairpins [83].

An ORI was also reported inside the *p143* gene of AcMNPV. This sequence contains 13 imperfect palindromic sequences of 12 to 18 bp in size [35]. Furthermore, the intergenic and intragenic fragments identified as ORI in our work are located inside or close to core genes, the common ancestral set of sequences conserved by all baculoviruses. This proximity could ensure their transfer to progeny and thus be sustained through the evolution, allowing them to conserve at least one mechanism of replication, which is still functional in all species. The presence of many ORIs around the genome could provide a redundancy required to ensure DNA replication. In vivo evidence supports the hypothesis that AcMNPV replication involves multiple ORIs that are activated with different efficiency during the viral infection cycle [42]. According to our results, the same could occur in AgMNPV.

The fact of having found and predicted a shared structure in all the sequences that act as ORI in baculoviruses, both those found by this work in AgMNPV and those found by other authors in other species, is in itself very relevant because such types of structures are usually key factors in replicative processes in other biological systems [83]. For example, the phage G4 (an ssDNA virus from *Escherichia coli*) carries three hairpins with stems of 5 to 19 bp and loops of 4 to 8 bases in the ORI of replication, which seems to direct the binding of single-stranded binding proteins [84]. The dsDNA bacterial plasmid R6K replicates by the theta mechanism and a DNA hairpin named M13-A is the core of the priming initiation [85]. In the plasmid RSF1010, which replicates by strand displacement, RepC protein binds to iterons and unwinds the dsDNA in a region that folds into hairpins [86]. In turn, the plasmid pT181 replicates by the rolling circle and contains a hairpin with the Rep nicking site, which initiates replication [87]. Thus, the structures that we have predicted in baculoviral ORIs are frequent in nature and are associated with replication functions under various mechanisms compatible with those that could occur in baculoviruses. Consequently, they would constitute the essential characteristic that requires a baculoviral sequence to be the initiator of the DNA synthesis process.

## Figures and Tables

**Figure 1 viruses-11-00648-f001:**
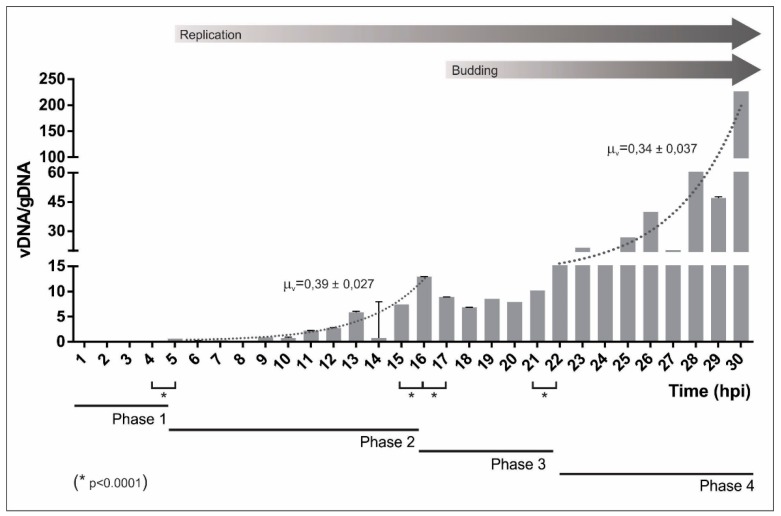
Replication kinetics of AgMNPV. Synchronized monolayers (70% confluence) of UFL-Ag-286 cells growing on 24 multiwell plates were exposed with AgMNPV (moi 10) for 1 h. Then, cells were incubated with growth medium for 1 to 30 h at 27 °C (hpi: hours post infection). DNA was recovered from the conditioned medium and cells by triplicate for each point. To quantify relative levels between cell and virus DNAs (gDNA and vDNA, respectively), qRT-PCR assays were applied. The results were plotted as quotients between vDNA and gDNA. The statistical error was calculated and then propagated to obtain the error bars. The beginning of replication and budding are indicated with grey arrows. Replication stages: Phase one, from 0 to 4 h; phase two, from 5 to 16 h; phase three, from 17 to 21 h; and phase four, from 22 to at least 30 h. The replication rates corresponding to phase two and four are indicated with a dotted line. An unpaired parametric *t*-test was done between the vDNA/gDNA values corresponding to contiguous times. The statistically significant differences (* *p* < 0.0001) that separate the replicative phases are indicated.

**Figure 2 viruses-11-00648-f002:**
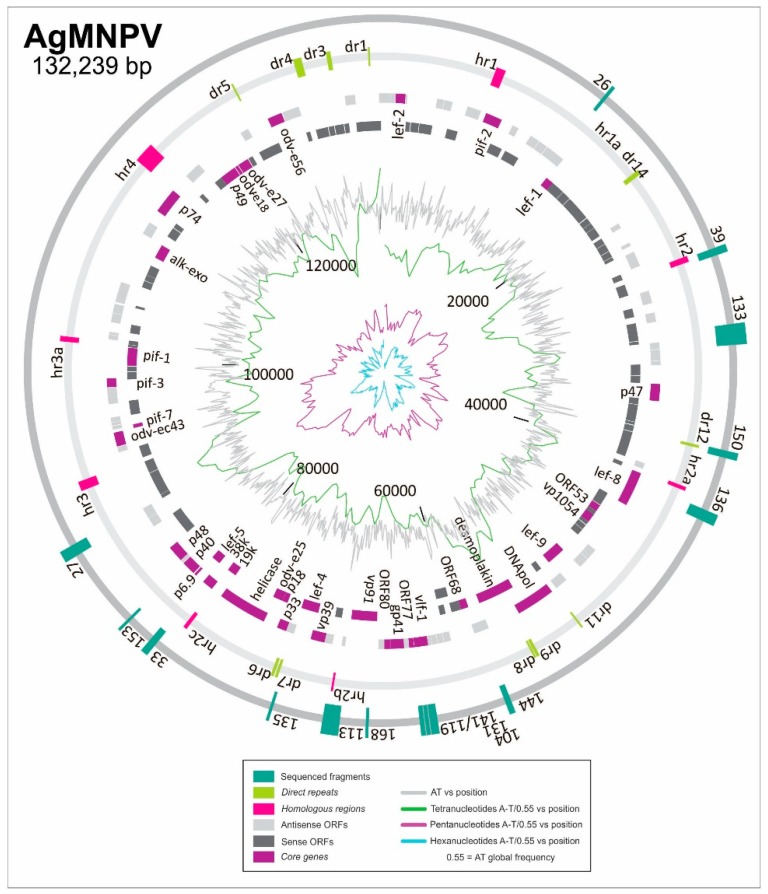
AgMNPV ORI sequences. The illustration shows the physical maps of AgMNPV loci containing the segments recovered as ORI by the transient replication assay. They are indicated as green boxes and named using the corresponding clone number derived from the AgMNPV library. The annotated ORFs are represented with different grey shaded boxes (dark grey and light grey). Core genes are indicated in violet boxes and named according to the traditional denomination. The homologous regions (*hrs*) and direct repetitions (*drs*) are indicated using pink and light green boxes, respectively. Inside the circle that represents the genome, the AT content (gray line) and the profiles of appearance of 4mer (green line), 5mer (violet line), and 6mer (blue line) runs containing only A or T are shown.

**Figure 3 viruses-11-00648-f003:**
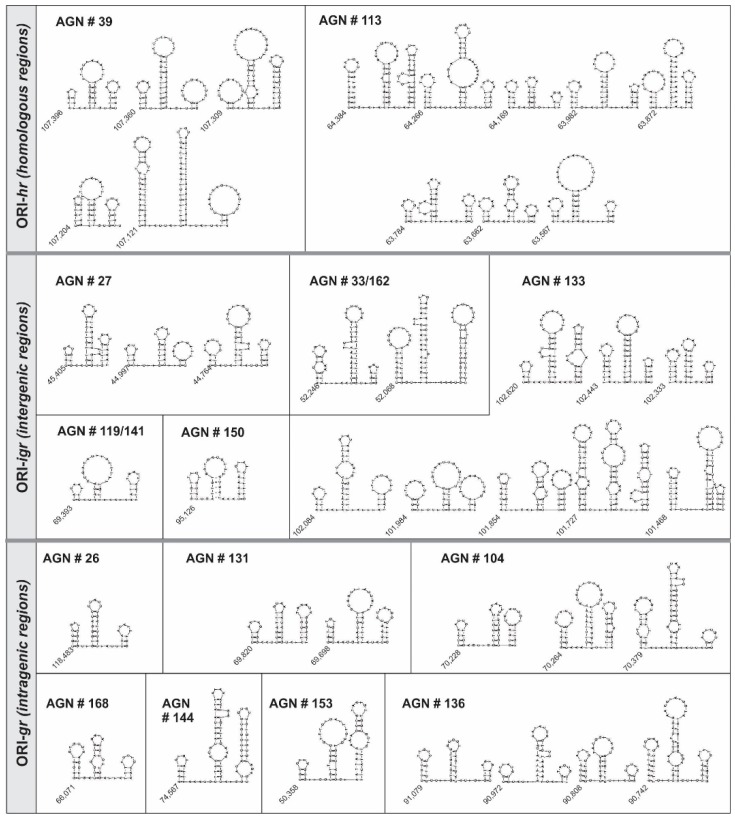
ORI sequences of AgMNPV and their putative structure. The illustration shows the putative secondary structures of the sequences recovered as ORI by the transient replication assay. ORIs are classified according their genome location and library clones are identified with their corresponding numbers. The nucleotide genome position of each sequence is indicated at the beginning of the DNA structure.

**Figure 4 viruses-11-00648-f004:**
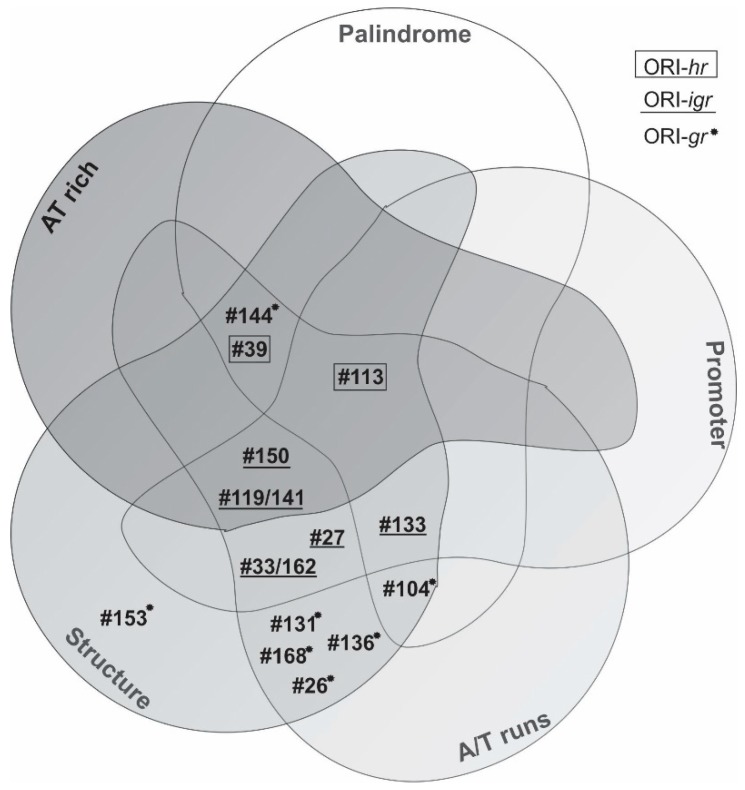
AgMNPV ORI sequences and their shared properties. Venn diagram shows shared properties among different ORIs of AgMNPV. The AT percentage has been selected only when values were over the average of the AgMNPV genome. Early promoters were considered functional when they are located close to the first ATG codon. In this graphic, only imperfect palindrome >10 nucleotides were considered.

**Table 1 viruses-11-00648-t001:** Sequences containing ORIs of AgMNPV.

**ORI-*hr***	**Clone Number**	**Position (bp)**	**Identity Sequence**				**Repetitions**			
	39	107,445–106,982	*hr*9				8× GCTTTWCRARYACMRTYRWTHKTGWAAARCD			
	113	64,519–63,441	*hr*7				1× GTTTTACAAAAACAATCGTATGTGAAAAAC			
**ORI-*Igr***	**Number**	**Position**	**Surrounding ORFs**				**Shared by**			
	27	45,424–44,663	ORF60 (*he65-like*), ORF58		ORF57 (*pnk-pnl*)		Most GI, some GII and 3 *Beta*		5 baculovirus genomes	
	119/141	69,305–69,564	ORF85 (*ac78-like)*		ORF86 (*vlf1*)		*Baculoviridae*		*Baculoviridae*	
	133	102,699–101,360	ORF126	ORF125	ORF124 (*lef-6*)	ORF123 (*iap-1*)	GI	GI, most GII and *Beta*	*Alpha* and *Beta*	GI
	150	95,156–94,754	ORF115		ORF114 (*odv-e66*)		GI and SeMNPV		GI, most GII, *Beta*	
	33/162	52,036–52,450	ORF67 (*lef-5*)		ORF68 (*38k*)		*Baculoviridae*		*Baculoviridae*	
**ORI-*gr***	**Number**	**Position**	**ORF**				**Shared by**			
	26	118,470–118,641	*egt*				GI, GII, and most *Beta*			
	104	69,883–70,458	*vlf-1*				*Baculoviridae*			
	131	60,314–60,131	*vlf-1*				*Baculoviridae*			
	136	90,683–91,393	*lef-8*				*Baculoviridae*			
	144	74,559–74,844	*lef-3*				*Alpha* and *Beta*			
	153	50,359–50,236	*p40*				*Baculoviridae*			
	168	66,087–65,983	*vp91*				*Baculoviridae*			

The clone number column refers to the AgMNPV library clones selected for its ability to act as ORI. ORI-hr (sequence containing homologous regions). ORI-Igr (sequence containing some partial or complete ORFs, including some intergenic region). ORI-gr (sequence immersed into an ORF). “Position” refers to the nucleotide position in the AgMNPV genome (NC_008520.2). Baculovirus classification clades are indicated as follows: GI (group I alphabaculoviruses), GII (group II alphabaculoviruses), alpha (alphabaculoviruses), beta (betabaculoviruses), *Baculoviridae* (all members sequenced of family *Baculoviridae*).

**Table 2 viruses-11-00648-t002:** Palindromes and promoters contained in ORIs of AgMNPV.

ORI	Clone Number		Promoter Pattern								Palindrome Pattern										
		Length (bp)	Early		Late		INR		TATAAA		Perfect					Imperfect					
			+	-	+	-	+	-	+	-	5 nt	6 nt	7 nt	8 nt	11 nt	5 nt	6 nt	7 nt	8 nt	9 nt	>10 nt
**ORI-*hr***	39	833	7	10	1	-	-	-	-	-	11	4	2	-	-	90	23	10	4	5	2
	113	1080	9 (3*)	11 (3*)	-	-	1*	1	-	-	30	13	2	-	1	147	43	25	10	3	2
**ORI-*igr***	27	763	4*	3*	-	-	-	-	-	1*	17	4	1	-	-	88	32	10	4	-	-
	119/141	260	3	1	1*	-	-	-	-	-	5	1	-	-	-	21	10	3	2	-	-
	133	1337	11 (1*)	11 (2*)	3 (1*)	-	-	-	-	-	28	9	3	1	-	150	68	13	4	2	2
	150	403	5 (2*)	-	1*	-	1*	-	1*	1	14	1	2	1	-	46	19	16	1	3	2
	33/162	415	2 (1*)	3*	-	-	1*	-	-	-	10	-	1	-	-	45	18	4	3	1	-
**ORI-*gr***	26	172	2	2	-	-	-	-	-	-	2	-	-	-	-	13	5	2	1	1	-
	104	576	1	4	-	-	1	-	-	-	6	-	1	1	-	57	31	8	-	1	1
	131	318	1	1	1	-	-	-	-	-	9	-	-	-	-	39	10	8	1	-	-
	136	711	7	10	1	2	-	1	-	-	13	-	5	1	-	69	30	6	3	-	-
	144	286	2	5	-	-	-	-	-	-	7	-	4	3	-	45	14	8	2	1	-
	153	124	-	2	-	-	-	-	-	-	2	-	-	-	-	6	2	-	1	-	-
	168	105	-	2	-	-	-		-	-	2	-	-	-	-	5	1	1	-	-	-

The clone number column refers to the AgMNPV library clones selected for its ability to act as ORI. ORI-hr (sequence containing homologous regions). ORI-Igr (sequence containing some partial or complete ORFs, including some intergenic region). ORI-gr (sequence immersed into an ORF). Plus (“+” in Promoter Pattern column) or minus (“-” in Promoter Pattern column) refers to the strand of dsDNA (considering the plus strand the sequence published on NC_008520.2). *Promoters that could be functional by its position close to one ORF. INR: initiator motif.

**Table 3 viruses-11-00648-t003:** Putative DNA structures of AgMNPV ORIs.

	Clone Number	Number of Structures	Start (bp)	Length					AT%					End (bp)
				S1 (bp)	Linker 1 (nt)	S2 (bp)	Linker 2 (nt)	S3 (bp)	S1	Linker 1	S2	Linker 2	S3	
**ORI-*hr***	**39**	5	107,396	3	2	6	2	4	77.7	100	65.2	0	53.8	107,348
			107,360	4	2	10	4	2	53.8	50	56.2	50	50	107,294
			107,309	2	2	8	3	8	50	100	60.5	33.3	56.5	107,227
			107,204	5	0	6	2	4	61.5	0	65.2	0	53.8	107,151
			107,121	14	6	15	6	3	50	66.7	62.5	66.7	45.4	107,010
	**113**	8	64,384	6	5	9	3	10	40.9	80	36.7	66.7	40	64,295
			64,266	5	5	10	3	4	81.2	80	72.5	0	76.9	64,187
			64,169	4	2	5	3	2	100	50	57.1	66.7	75	64,130
			63,982	4	4	8	4	4	84.6	50	96.1	100	63.6	63,923
			63,872	4	2	11	1	6	63.2	100	96.8	100	88.2	63,803
			63,784	3	3	7	5	4	54.5	66.7	65.2	80	76.9	63,730
			63,662	3	3	6	2	2	83.3	100	52.4	0	55.6	63,616
			63,567	3	2	7	5	3	66.7	100	73.5	80	40.0	63,505
**ORI-*Igr***	**27**	3	45,405	3	2	10	1	5	33.3	100.0	63.0	100.0	64.3	45,353
			44,997	3	4	6	2	2	50.0	75.0	70.6	100.0	53.8	44,952
			44,764	3	3	8	3	4	53.8	66.7	65.5	33.3	66.7	44,709
	**119/141**	1	69,393	4	1	9	3	6	92.3	100.0	69.7	66.7	47.4	69,464
	**133**	8	102,620	3	3	10	3	8	60.0	66.7	44.1	66.7	33.3	102,547
			102,443	6	2	8	2	4	64.7	100.0	43.3	0.0	45.5	102,382
			102,333	5	1	7	2	3	86.7	0.0	63.2	50.0	60.0	102,287
			102,084	3	3	11	5	4	90.9	100.0	81.3	100.0	82.4	102,017
			101,984	3	4	5	3	3	40.0	50.0	45.8	33.3	61.1	101,921
			101,854	6	5	6	0	5	56.3	80.0	40.9	0	36.8	101,793
			101,727	12	4	11	4	10	51.4	75.0	42.5	75.0	36.7	101,619
			101,468	7	5	11	0	4	77.8	60.0	44.4	0	27.3	101,394
	**150**	1	95,126	4	2	5	3	6	50.0	0.0	70.0	100.0	75.0	95,074
	**33/162**	2	52,068	5	5	13	2	3	56	100	75	50.0	44	52,137
			52,246	8	3	16	6	10	66.7	33.3	71.1	83.3	41.7	52,355
**ORI-*gr***	**26**	1	118,483	4	2	7	4	3	18.2	50.0	28.6	50.0	54.5	118,531
	**104**	3	70,228	4	5	6	0	5	41.7	40	47.4	0	33.3	70,281
			70,264	5	3	8	2	7	33.3	100	64.5	50	45	70,337
			70,379	6	4	13	5	2	65.2	75.0	30	60.0	30.0	70,457
	**131**	2	69,698	3	3	5	3	5	0.8	66.7	39	66.7	66.7	69,750
			69,820	4	4	6	3	5	36.4	50	66.7	66.7	52.9	69,881
	**136**	4	90,742	7	3	11	4	5	70	66.7	51.3	25	46.7	90,822
			90,808	5	2	6	4	2	47	50.0	40.9	75	22.2	90,859
			90,972	2	5	9	3	2	40	60	67	66.7	44	91,025
			91,079	5	4	6	5	3	20	50	37	60.0	70	91,131
	**144**	1	74,567	4	5	14	3	10	83.3	60.0	75.0	66.7	16.1	74,657
	**153**	1	50,358	3	4	7	3	4	30.0	50.0	41.4	66.7	82.4	50,296
	**168**	1	66,071	4	2	6	4	3	37.5	50.0	36.8	100.0	72.7	66,020

The clone number column refers to the AgMNPV library clones selected for its ability to act as ORI. ORI-hr (sequence containing homologous regions). ORI-Igr (sequence containing some partial or complete ORFs, including some intergenic region). ORI-gr (sequence immersed into an ORF). The common structure found in all the AgMNPV ORIs is described in this table, indicating the first and last nucleotide involved (refers as Start and End columns) and the length and AT content of each component (S1: stem 1; linker 1; S2: stem 2; linker 2; S3: stem 3). Different rows for each ORI in length and AT% columns contained data from each of the predicted structures.

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
