# Peer review of "Identification of Multiple Replication Stages and Origins in the Nucleopolyhedrovirus of Anticarsia gemmatalis"

_viruses, 2019, doi:10.3390/v11070648_

Round 1
Reviewer 1 Report
For the Ori cloning and functional enrichment experiment it would be helpful to have a flow diagram depicting how the pieces containing and not containing Ori sequences make it through the protocol to be identified as containing and Ori.. There are aspects of the HaeIII partial vs overnight digests that are not clear.
Specific English changes recommended..Line:
109) insert "post infection" before into the susceptible and replace "during " with "hourly for"
130) insert "our" before own data or replace own with "Data obtained in this lab ( and a accession number)
149) replace "during" with "for"
153) "On the other hand" should be replaced with "For the remainder of the sample" DNA was...
156) NaOAc is a more standard abbreviation for Sodium Acetate.
163) replace "stastistic" with "statistical" and ratios "were" calculated..
195) "As a control" should be moved to the beginning of the sentence and replace "Besides"
197) replace "Exposition" with "exposure" or "infection"
198) replace "After this time" with "At 48 hrs"
199) replace "Then" with "The"
203) Is the :"Previous certification of control samples" supposed to be a header like Bioinformatics analysis ? "To this" should be "The"
233) insert "analysis" after kinetic
243) replace "exposition" with "exposure" or "treatment"
250) replace "since" with "from" or "between"
300) "viral virus" Choose one!
379) "on the other hand" is overused
431) replace " informed" with "reported"
432) "Besides" is unnecessary
433) insert "," between predicted and thinking
Table 2) replace "lenght" with Length" also there are several places where the text wrapping functions are making it difficult to follow.
456) replace "In view that" with "As"
491) replace "analyzes" with "analyses" which is the plural of analysis
587) "On the other hand " is overused and unnecessary
Table S1) text wrapping issues
Author Response
Thank you very much for the comments and suggestions. I hope you are convinced that we have adequately addressed your comments.
For the Ori cloning and functional enrichment experiment it would be helpful to have a flow diagram depicting how the pieces containing and not containing Ori sequences make it through the protocol to be identified as containing and Ori. There are aspects of the HaeIII partial vs overnight digests that are not clear.
We have added a supplementary figure that shows the workflow associated with the identification of the replication ORI. We have also clarified some aspects associated with this procedure in the manuscript.
Specific English changes recommended. .Line:
We appreciate the careful work of the reviewer. All the observations are relevant, and we have introduced them in the manuscript using the "Track Changes" function in Microsoft Word. We have also improved the organization of the tables without applying "Track Changes" because this function makes very difficult the proper visualization of the tables.
109) insert "post infection" before into the susceptible and replace "during " with "hourly for"
We agree with Reviewer 1 and we changed the manuscript accordingly in line 111.
130) insert "our" before own data or replace own with "Data obtained in this lab ( and a accession number)
We added the GenBank ID (KX139200.1) in line 132 that contains the partial cds of Anticarsia gemmatalis beta-actin gene.
149) replace "during" with "for"
Corrected, line 150.
153) "On the other hand" should be replaced with "For the remainder of the sample" DNA was...
Corrected, line 155.
156) NaOAc is a more standard abbreviation for Sodium Acetate.
We agree and the changed the abbreviation, line 158.
163) replace "stastistic" with "statistical" and ratios "were" calculated.
Corrected, lines 164.
195) "As a control" should be moved to the beginning of the sentence and replace "Besides"
Corrected, line 195.
197) replace "Exposition" with "exposure" or "infection"
Corrected, line 197.
198) replace "After this time" with "At 48 hrs ok
Corrected, line 198.
199) replace "Then" with "The"
203) Is the :"Previous certification of control samples" supposed to be a header like Bioinformatics analysis ? "To this" should be "The"
We reorganized this paragraph, from line 200 to 216.
233) insert "analysis" after kinetic
Corrected, line 238.
243) replace "exposition" with "exposure" or "treatment"
Corrected, line 249.
250) replace "since" with "from" or "between"
Corrected, line 256.
300) "viral virus" Choose one!
Corrected, line 305.
379) "on the other hand" is overused
Corrected, line 384.
431) replace " informed" with "reported"
Corrected, line 436.
432) "Besides" is unnecessary
Corrected, line 437.
433) insert "," between predicted and thinking
Corrected, line 438.
Table 2) replace "lenght" with Length" also there are several places where the text wrapping functions are making it difficult to follow.
Corrected.
456) replace "In view that" with "As"
Corrected, line 462.
491) replace "analyzes" with "analyses" which is the plural of analysis
Corrected, line 496.
587) "On the other hand " is overused and unnecessary
Corrected, line 592.
Table S1) text wrapping issues
Corrected.

Reviewer 2 Report
In the manuscript “Identification of multiple replication stages and origins in the nucleopolyhedrovirus of Anticarsia gemmatalis”, Miele et al. identify 12 virus fragments that acted as ORI during baculovirus infection. In their report, the authors have illustrated common structures on those fragments that probably recruit the replication machinery and have described their methods and results in sufficient detail. However, the mechanism of replication used by baculoviruses was not studied nor did the authors studied the factors (including virus and host proteins) involved in viral DNA replication. Also, the sequences crucial for DNA replication and the sequences where DNA synthesis begins were not studied or identified. Thus, fundamentally, this study simply accumulates data derived from sequencing of DNA replications during different stages of baculovirus infection, but lacks crucial information necessary to advance our knowledge regarding baculovirus DNA replication.
Major comments
1. In Figs. 1 and 2, although the authors found and classified 14 genomic ORI regions into three main groups according to their locations—ORI-hr; ORI-igr, and ORI-gr—they should provide a table to state which ORI is replicated during different stages of virus infection. In addition, they should compare the sequence differences between the putative ORIs during early and late stages, and summarize their conclusions.
2. The authors discussed interactions between IE-1 protein and ORIs. It would be helpful to predict or study the interactions of IE-1 with the secondary structures they have elaborated for these ORIs.
3. The authors have also plotted where DNA is replicated in the genomic map and indicated the palindromes and sets of three hairpin units in the putative ORIs, but they do not provide any data about how these hairpins work nor give information relating to the core or essential ORI sequences or the mechanism of DNA replication.
Minor comment:
Table 2, abbreviations are not clearly stated. It takes time to determine Plu=Plus strand and Min=Minus strand. Moreover, “s” and “us” are not explained.
Author Response
Thank you very much for the comments and suggestions. I hope you are convinced that we have adequately addressed your comments.
In the manuscript “Identification of multiple replication stages and origins in the nucleopolyhedrovirus of Anticarsia gemmatalis”, Miele et al. identify 12 virus fragments that acted as ORI during baculovirus infection. In their report, the authors have illustrated common structures on those fragments that probably recruit the replication machinery and have described their methods and results in sufficient detail. However, the mechanism of replication used by baculoviruses was not studied nor did the authors studied the factors (including virus and host proteins) involved in viral DNA replication. Also, the sequences crucial for DNA replication and the sequences where DNA synthesis begins were not studied or identified. Thus, fundamentally, this study simply accumulates data derived from sequencing of DNA replications during different stages of baculovirus infection, but lacks crucial information necessary to advance our knowledge regarding baculovirus DNA replication.
We understand the opinions of the reviewer, but although our work does not provide clear evidence on the identification of the baculovirus´ replication mechanism(s), it does provide for the first time a detailed description of viral DNA synthesis progression throughout the infection cycle, including the measurement of replication rates. We show with statistical significance the presence of 4 different stages associated with the production of viral DNA.
Regarding the genomic sequences identified containing potential ORI of replication, we would like to point the fact that we included the entire genome. Under this approach, we found sequences previously described as origins of replication in other baculovirus species, giving consistency to our results. We analyzed those sequences in many ways and we found they could fold in a defined way. We performed a stringent bioinformatics analysis to find those structures, as detailed in materials and methods and the tables showing the stem and linker length. Moreover, we observed that the putative ORI sequences localize in genomic regions that are conserved through evolution, meaning the essential genome not only contains the essential coding regions, but also would contain important non-coding sequences relevant for the multiplication cycle.
To our knowledge, our work gathers important information to delve into the replication process of baculoviruses. There is no previous publication showing different stages during viral DNA replication. There are publications about hr, non-hr and early promoters as origins, but none of them concludes about a specific mechanism. Our work is a step forward in many years of studies about baculoviral replication and will collaborate with future studies to explore in a more precise way which factors are involved in each stage.
Major comments
1. In Figs. 1 and 2, although the authors found and classified 14 genomic ORI regions into three main groups according to their locations—ORI-hr; ORI-igr, and ORI-gr—they should provide a table to state which ORI is replicated during different stages of virus infection. In addition, they should compare the sequence differences between the putative ORIs during early and late stages and summarize their conclusions.
Based on previous reports, we could not conclude neither which sequence is the origin of replication or the mechanism used by baculoviruses. From there, our study represents a first stage in a broader exploration of the replication process of alphabaculoviruses. Therefore, the methodological approach aimed to obtain sequences that act as ORI, without any bias. We wanted to find sequences that can recruit the replication machinery throughout the whole the infective process, and this is the reason to include in this work an experiment done for 48 hours of incubation, that includes all the replicative phases that we also describe. Since the sequences previously described and those found in this work have common characteristics, we consider they likely participate in the same mechanism and therefore might be associated to a single phase.
In agreement with Reviewer 2, we consider it would be very useful to show if the different ORIs are associated with the different stages of replication, however this information is not available yet, but we are working on this line.
2. The authors discussed interactions between IE-1 protein and ORIs. It would be helpful to predict or study the interactions of IE-1 with the secondary structures they have elaborated for these ORIs.
Although IE-1 was originally characterized as a transcriptional activator, there is a large amount of literature that demonstrates that it is essential for viral replication through its binding to hr sequences. Even, it has been shown that IE-1 localizes to the virogenic stroma [1]. This protein has different domains, including two acidic activation domains and one basic domain (basic domain I) at the N terminus [2]; an additional basic domain (basic domain II) and a helix-loop-helix domain at the C terminus. The basic domain I is required for hr binding (sequence-specific DNA binding domain) [3], while the domains in the C terminus are involved in DNA binding, oligomerization and nuclear import. On the other hand, the existence of a replication domain in the first 60 residues of the N terminus of IE-1 has also been proposed [4]. IE-1 bound to DNA after forming a homodimeric complex [5]. Moreover, oligomerization is a prerequisite for its import into the nucleus [6]. Essentially, any of these known DNA binding domains could interact with the nucleotides of the ORI once they acquire the predicted secondary structure. A possible bioinformatic analysis to predict this interaction could be to generate a model of the three-dimensional structure of IE-1 homodimer. Then, using this 3D structure and an oligonucleotide with the structure detected as conserved in the possible ORI, we could carry out analysis of Molecular Docking and Molecular Dynamics. These studies would allow us to determine if IE-1 homodimer could interact with the predicted structure or not, allowing us to even predict which amino acids are involved in the interaction. However, to construct this model no protein structure is available. Moreover, purification of biologically active IE-1 has not been reported suggesting that it might be unstable or only active in combination with another protein(s) [7]. By homology comparison of IE-1 in Swiss-Model (https://swissmodel.expasy.org) [8], we obtained results with a very low coverage, making it difficult to carry out a model of this protein. A model can be done, however the biological significance of the model obtained can be doubted due to the low similarity of IE-1 to the proteins in the databases.
3. The authors have also plotted where DNA is replicated in the genomic map and indicated the palindromes and sets of three hairpin units in the putative ORIs, but they do not provide any data about how these hairpins work nor give information relating to the core or essential ORI sequences or the mechanism of DNA replication.
To our knowledge, there were no consensus about which sequences are the ORI of replication in baculovirus. As we mentioned in the Introduction of our manuscript, there are plenty of work done regarding this subject, however the question still remains. Although hr and non-hr ORI sequences have been examined [hr: 1,2; non-hr: 3–5; early promoters: 6], it is still not clear which, if any, of these sequences is the origin of replication in vivo [7].
We attempted to confirm if those reported sequences could function as ORI, and also found all the possible sequences having this function. The mechanism behind these sequences is another great question in which we are working right now. However, the answer is not easy since multiple proteins and factors are involved. Some of those proteins are essential for replication, and cannot be removed, or are difficult to purify to test in vitro.
Regarding the possible structure found in the reported sequences, the phage G4, a ssDNA phage from Escherichia coli, carries three hairpins with stems of 5 to 19 bp and loops of 4 to 8 bases in the ORI of replication region, that seems to direct the binding of SSB (Single-Stranded Binding Protein) so that the primase recognition site, exposed by this structure [15].
In the case of dsDNA, it has been proposed that cruciform structures could be present in ORI of replication [16]. These structures have been shown in vitro [17], but their formation in vivo is difficult to assess. During replication, strand separation by the helicase leads to the positive supercoiling of the duplex ahead of the fork [18]; this could favor the acquisition of superhelical densities needed to favor the formation of cruciform structures. It has been proposed that hrs could adopt a cruciform structure, but IE-1 can bind the sequence when it forms a hairpin and when it doesn’t [19].
We transcribe a very interesting paragraph from [21]:
“In bacteria, replication of dsDNA starts with the melting of a region of DNA, which is favoured by a protein complex, that binds the DNA and bends it [20]. This bending promotes also formation of alternative DNA structures [16], so the same binding of proteins could change DNA conformation. In eukaryotes, cruciform structures are targets for many architectural and regulatory proteins, such as histones H1 and H5, topoisomerase IIβ, HMG proteins, HU, p53, the proto-oncogene protein DEK and others. Several DNA-binding proteins, such as the HMGB-box family members, Rad54, BRCA1 protein, as well as PARP-1 polymerase, possess weak sequence specific DNA binding yet bind preferentially to cruciform structures. Some of these proteins are, in fact, capable of inducing the formation of cruciform structures upon DNA binding.”
It has been shown that hairpins play essential roles in primosome assembly in dsDNA replication. The generation of a primer occurs by the opening of the DNA double helix followed by RNA priming (chromosomal, theta, and strand displacement replications) [22] or the cleavage of one of the DNA strands to generate a 3-OH end (Rolling Circle Replication) [23]. For both mechanisms, there are cases where hairpins are involved. For example, in R6K plasmids that replicate by theta mechanism, a hairpin named M13-Ais is the core of priming mechanism [24]. For plasmid RSF1010 that replicates by strand displacement, RepC protein binds to iterons and unwinds the DNA in a region that folds into hairpins [25]. The plasmid pT181 replicates by rolling circle, and contains a harpin with the Rep nicking site, that initiates replication [26].
Hairpins can be involved in many mechanisms in agreement with everything commented previously, and we are working to finally assess the mechanism used by baculoviruses. However, many groups have already tried, and they found it is a difficult task. It has been proposed that baculovirus replicate by rolling circle [27]; however, a homologous to Rep has not been detected. Also, recombination-dependent replication has been proposed as a mechanism [7], but the stability of baculovirus genomes is against this hypothesis.
We consider that the fact of having found and predicted a shared structure in all the sequences that act as ORI in baculovirus, both those found by us and those found by other authors in other species, is in itself very relevant because such type of structures are usually key factors in replicative processes as we mentioned before. This evidence provided by our work can help other researchers to design appropriate experiments to determine the role of these genome structures during the replication process.
References used in our responses to Reviewer 2:
1. Kawasaki, Y.; Matsumoto, S.; Nagamine, T. Analysis of baculovirus IE1 in living cells: dynamics and spatial relationships to viral structural proteins. J. Gen. Virol. 2004, 85, 3575–83.
2. Dai, X.; Willis, L.G.; Huijskens, I.; Palli, S.R.; Theilmann, D.A. The acidic activation domains of the baculovirus transactivators IE1 and IE0 are functional for transcriptional activation in both insect and mammalian cells. J. Gen. Virol. 2004, 85, 573–582.
3. Olson, V.A.; Wetter, J.A.; Friesen, P.D. The Highly Conserved Basic Domain I of Baculovirus IE1 Is Required for hr Enhancer DNA Binding and hr-Dependent Transactivation. J. Virol. 2003, 77, 5668–5677.
4. Taggart, D.J.; Mitchell, J.K.; Friesen, P.D. A conserved N-terminal domain mediates required DNA replication activities and phosphorylation of the transcriptional activator IE1 of Autographa californica multicapsid nucleopolyhedrovirus. J. Virol. 2012, 86, 6575–85.
5. Rodems, S.M.; Friesen, P.D. Transcriptional enhancer activity of hr5 requires dual-palindrome half sites that mediate binding of a dimeric form of the baculovirus transregulator IE1. J. Virol. 1995, 69, 5368–75.
6. Olson, V.A.; Wetter, J.A.; Friesen, P.D. Baculovirus transregulator IE1 requires a dimeric nuclear localization element for nuclear import and promoter activation. J. Virol. 2002, 76, 9505–15.
7. F, R.G. Baculovirus Molecular Biology; 3rd ed.; Bethesda (MD): National Center for Biotechnology Information (US), 2013;
8. Waterhouse, A.; Bertoni, M.; Bienert, S.; Studer, G.; Tauriello, G.; Gumienny, R.; Heer, F.T.; de Beer, T.A.P.; Rempfer, C.; Bordoli, L.; et al. SWISS-MODEL: homology modelling of protein structures and complexes. Nucleic Acids Res. 2018, 46, W296–W303.
9. Pearson, M.; Bjornson, R.; Pearson, G.; Rohrmann, G. Science. Science (80-. ). 1992, 209, 1392–1396.
10. Kool, M.; van den Berg, P.M.M.M.; Tramper, J.; Goldbach, R.W.; Vlak, J.M. Location of Two Putative Origins of DNA Replication of Autographa californica Nuclear Polyhedrosis Virus. Virology 1993, 192, 94–101.
11. Lee, H.Y.; Krell, P.J. Generation and analysis of defective genomes of Autographa californica nuclear polyhedrosis virus. J. Virol. 1992, 66, 4339–47.
12. Pearson, M.N.; Bjornson, R.M.; Ahrens, C.; Rohrmann, G.F. Identification and characterization of a putative origin of DNA replication in the genome of a baculovirus pathogenic for Orgyia pseudotsugata. Virology 1993, 197, 715–25.
13. Habib, S.; Hasnain, S.E. Differential activity of two non-hr origins during replication of the baculovirus Autographa californica nuclear polyhedrosis virus genome. J. Virol. 2000, 74, 5182–9.
14. Wu, Y.; Liu, G.; Carstens, E.B. Replication, integration, and packaging of plasmid DNA following cotransfection with baculovirus viral DNA. J. Virol. 1999, 73, 5473–80.
15. Sun, W.; Godson, G.N. Structure of the Escherichia coli primase/single-strand DNA-binding protein/phage G4oric complex required for primer RNA synthesis. J. Mol. Biol. 1998, 276, 689–703.
16. Bikard, D.; Loot, C.; Baharoglu, Z.; Mazel, D. Folded DNA in Action: Hairpin Formation and Biological Functions in Prokaryotes. Microbiol. Mol. Biol. Rev. 2010, 74, 570–588.
17. Murchie, A.I.H.; Lilley, D.M.J. The mechanism of cruciform formation in supercoiled DNA: initial opening of central basepairs in salt-dependent extrusion. Nucleic Acids Res. 1987, 15, 9641–9654.
18. Schvartzman, J.B.; Stasiak, A. A topological view of the replicon. EMBO Rep. 2004, 5, 256–61.
19. Rasmussen, C.; Leisy, D.J.; Ho, P.S.; Rohrmann, G.F. Structure-function analysis of the Autographa californica multinucleocapsid nuclear polyhedrosis virus homologous region palindromes. Virology 1996, 224, 235–45.
20. Mott, M.L.; Berger, J.M. DNA replication initiation: mechanisms and regulation in bacteria. Nat. Rev. Microbiol. 2007, 5, 343–354.
21. Brázda, V.; Laister, R.C.; Jagelská, E.B.; Arrowsmith, C. Cruciform structures are a common DNA feature important for regulating biological processes. BMC Mol. Biol. 2011, 12, 33.
22. del Solar, G.; Giraldo, R.; Ruiz-Echevarría, M.J.; Espinosa, M.; Díaz-Orejas, R. Replication and control of circular bacterial plasmids. Microbiol. Mol. Biol. Rev. 1998, 62, 434–64.
23. Khan, S.A. Plasmid rolling-circle replication: highlights of two decades of research. Plasmid 2005, 53, 126–36.
24. Masai, H.; Nomura, N.; Arai, K. The ABC-primosome. A novel priming system employing dnaA, dnaB, dnaC, and primase on a hairpin containing a dnaA box sequence. J. Biol. Chem. 1990, 265, 15134–44.
25. Miao, D.M.; Honda, Y.; Tanaka, K.; Higashi, A.; Nakamura, T.; Taguchi, Y.; Sakai, H.; Komano, T.; Bagdasarian, M. A base-paired hairpin structure essential for the functional priming signal for DNA replication of the broad host range plasmid RSF1010. Nucleic Acids Res. 1993, 21, 4900–3.
26. Noirot, P.; Bargonetti, J.; Novick, R.P. Initiation of rolling-circle replication in pT181 plasmid: initiator protein enhances cruciform extrusion at the origin. Proc. Natl. Acad. Sci. U. S. A. 1990, 87, 8560–4.
27. Oppenheimer, D.I.; Volkman, L.E. Evidence for rolling circle replication of Autographa californica M nucleopolyhedrovirus genomic DNA. Arch. Virol. 1997, 142, 2107–13.
Minor comment:
Table 2, abbreviations are not clearly stated. It takes time to determine Plu=Plus strand and Min=Minus strand. Moreover, “s” and “us” are not explained.
We agree and we corrected as indicated by Reviewer 2. We apologize because due to the width of the columns in the sent manuscript, some words were fragmented, such as Plus and Minus.

Reviewer 3 Report
The present work demonstrates the identification of replication origins in the genome of AgMNPV via an unbiased approach through the transfection of a shotgun plasmid library into insect cells, followed by baculovirus infection, selection of replicated plasmids, and sequencing. This part of the study did not reveal novel or significant information, as it recapitulated what was already known from previous, similar studies on other baculoviruses. Briefly, HR and other repeat regions in the baculovirus genome were identified as being able to initiate replication, as has been shown before for other baculoviruses. While techically sound and thorough, it is of relatively little interest as nothing new or original is revealed. It would be of interest for example if the authors had taken the sequence and predicted structural information on AgMNPV origins and tested a synthetic origin containing these features for replication competence - this would provide experimental evidence that their structural and sequence observations are meaningful and not just coincidences.
The second part of this work quantifies replication kinetics of AgMNPV in a specific Ag cell line. Again, the work is technically sound, but doesn't reveal any particularly novel or interesting information - replication is slow at first, then viruses start budding and replication takes off, as it does with other baculoviruses. Here again, the work could have been made much more interesting by coupling it with the first part: which origins or which type of origins are involved in one or the other phases of replication.
In summary, whilst not particularly exciting to this reviewer, the work is thorough and technically sound, and the authors have done an excellent job in referencing the relevant literature and precedents, and have placed their results in the context of previously obtained results.
Overall, the manuscript would probably benefit from a proofreading by a native English speaker, as there are some minor English issues that should be corrected.
Please rephrase and clarify:
21: "In this work, we performed an AgMNPV DNA replication kinetics into the susceptible UFL-Ag-286 cells"
223: "A kinetic of virus DNA synthesis using in vitro cell cultures was performed to contribute with experimental evidence about AgMNPV genome´s replication"
Author Response
Thank you very much for the comments and suggestions. I hope you are convinced that we have adequately addressed your comments.
The present work demonstrates the identification of replication origins in the genome of AgMNPV via an unbiased approach through the transfection of a shotgun plasmid library into insect cells, followed by baculovirus infection, selection of replicated plasmids, and sequencing. This part of the study did not reveal novel or significant information, as it recapitulated what was already known from previous, similar studies on other baculoviruses. Briefly, HR and other repeat regions in the baculovirus genome were identified as being able to initiate replication, as has been shown before for other baculoviruses. While techically sound and thorough, it is of relatively little interest as nothing new or original is revealed. It would be of interest for example if the authors had taken the sequence and predicted structural information on AgMNPV origins and tested a synthetic origin containing these features for replication competence - this would provide experimental evidence that their structural and sequence observations are meaningful and not just coincidences.
Please see the answer given to reviewer 2 in the point 3 about the structure of the DNA found in ORI sequences.
We agree with the reviewer about the generation of a synthetic ORI that could fold as the predicted structure. We consider it would be of great value to highlight its role in baculoviral replication through the development of appropriate experimental setting. We are working on this direction, but we understand that in this first stage, our work is sufficient to provide valuable information on the replication of baculoviruses, among with the proposal of these structures as important in the synthesis of viral DNA, because they can be predicted in the sequences that we found and in those found by other authors in other species. We consider that our work, as it has been presented, is relevant and opens doors for future research on the field, including studies with putative synthetic ORIs.
The second part of this work quantifies replication kinetics of AgMNPV in a specific Ag cell line. Again, the work is technically sound, but doesn't reveal any particularly novel or interesting information - replication is slow at first, then viruses start budding and replication takes off, as it does with other baculoviruses. Here again, the work could have been made much more interesting by coupling it with the first part: which origins or which type of origins are involved in one or the other phases of replication.
There is no previous publication showing the different stages during baculoviral DNA replication, including the estimation of DNA synthesis rates. In this sense, our work provides relevant information to understand the replication mechanism (s), not only by us but for the entire baculoviral community. To our knowledge, the presented work gathers important information to delve into the replication process of baculoviruses.
Our study is a first stage in the exploration of the replication process of alphabaculoviruses, working based on everything previously reported, information that was profusely collected and included in the construction of our study. Therefore, in the methodological approach aimed to search sequences that act as ORI, we did not look for an association with a particular stage of the 2 exponential phases that we identified, but our objective was to find sequences that recruit the replication machinery throughout the whole process. Thus, we preferred to contain all the phases in the experiment (48 hours of incubation) in order to have the possibility of detecting all sequences that fulfilled such purposes (and to increase the sensitivity of the method). Subsequent bioinformatic characterization showed us that there are common features in these sequences, which may indicate that all the regions that we found, and that other authors found, participate in the same mechanism.
We consider it would be very useful to show if the different ORIs are associated with the different stages of replication, however this information is not available yet and we believe it will be important, but for future works.
In summary, whilst not particularly exciting to this reviewer, the work is thorough and technically sound, and the authors have done an excellent job in referencing the relevant literature and precedents and have placed their results in the context of previously obtained results.
We appreciate the reviewer's observation. In this sense, we strongly believe that our work contributes to the field, and that it provides relevant information to stimulate other authors to continue with expanding knowledge about the replication of these viruses.
Overall, the manuscript would probably benefit from a proofreading by a native English speaker, as there are some minor English issues that should be corrected.
We have reviewed the entire manuscript carefully.
Please rephrase and clarify:
21: "In this work, we performed an AgMNPV DNA replication kinetics into the susceptible UFL-Ag-286 cells"
We have introduced some changes that we hope will contribute to clarify the text.
223: "A kinetic of virus DNA synthesis using in vitro cell cultures was performed to contribute with experimental evidence about AgMNPV genome´s replication"
We have introduced some changes that we hope will contribute to clarify the text.
